# AutoDocSegmenter: A Geometric Approach towards Self-Supervised Document Segmentation

**Ankita Chatterjee**                                    *ankita.chatterjee25@kgpian.iitkgp.ac.in*
*Department of Computer Science and Engineering*
*Indian Institute of Technology Kharagpur*

**Anjali Raj**                                           *anjali.raj@kgpian.iitkgp.ac.in*
*Department of Computer Science and Engineering*
*Indian Institute of Technology Kharagpur*

**Soumyadeep Dey**                                       *desoumya@microsoft.com*
*Microsoft, INDIA*

**Pratik Jawanpuria**                                    *pratik.jawanpuria@microsoft.com*
*Microsoft, INDIA*

**Jayanta Mukherjee**                                    *jay@cse.iitkgp.ac.in*
*Department of Computer Science and Engineering*
*Indian Institute of Technology Kharagpur*

**Partha Pratim Das**                                    *ppd@ashoka.edu.in*
*Department of Computer Science*
*Ashoka University*

**Reviewed on OpenReview:** *https://openreview.net/forum?id=JBveijn200*

## Abstract

Document segmentation, the process of dividing a document into coherent and significant regions, plays a crucial role for diverse applications that require parsing, retrieval, and categorization. However, most existing methods rely on supervised learning, which requires large-scale labeled datasets that are costly and time-consuming to obtain. In this work, we propose a novel self-supervised framework for document segmentation that does not require labeled data. Our framework consists of two components: (1) an unsupervised isothetic covers based pseudo mask generator which approximately segments document objects, and (2) an encoder-decoder network that learns to refine the pseudo masks and segments the document objects accurately. Our approach can handle diverse and intricate document layouts by leveraging the rich information from unlabeled datasets. We demonstrate the effectiveness of our approach on several benchmarks, where it outperforms state-of-the-art document segmentation methods. Our code is available at https://github.com/ankitachatterjee94/AutoDocSegmenter

## 1 Introduction

Document digitization transforms paper-based documents into machine-readable formats that can be accessed and processed by various devices and applications. Document layout analysis (DLA) is a crucial step in digitization, as it aims to extract the information and structural components of a document, such as text, images, tables and graphs (Namboodiri & Jain, 2007; Lee et al., 2019). DLA enables key-value information extraction and localization, which are essential for tasks such as text recognition, retrieval, segmentation,

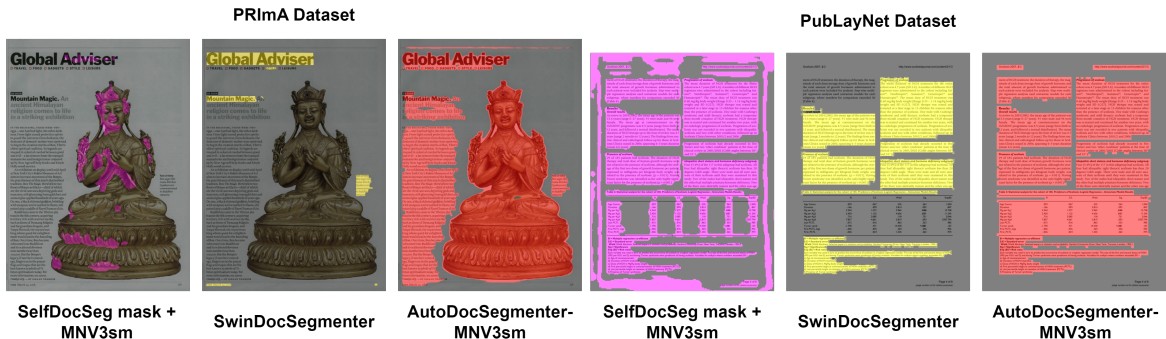

Figure 1: Segmenting typical document images from *PRImA* (Antonacopoulos et al., 2009) and *PubLayNet* (Zhong et al., 2019) datasets using *SelfDocSeg* mask (Maity et al., 2023) with MobileNetV3-small (Howard et al., 2019) backbone (MNV3sm), *SwinDocSegmenter* (Banerjee et al., 2023) and our proposed approach, AutoDocSegmenter-MNV3sm.

and classification. However, the performance of these tasks depends largely on the quality and type of annotations provided for the documents. For instance, while rectangular bounding boxes may suffice for text recognition and classification, but segmentation requires more accurate polygonal boundaries to capture the fine details of the document objects.

Existing works (Sun et al., 2019; Biswas et al., 2021) have explored *ConvNet*-based models such as Faster RCNN (Ren et al., 2015) and Mask RCNN (He et al., 2017b) for document object segmentation. More recently, transformer-based architectures have shown superior performance over *ConvNets* by exploiting global attention layers for document segmentation (Appalaraju et al., 2021; Huang et al., 2022). However, these supervised models rely heavily on the availability and quality of labeled data for training. Labeling documents with polygonal boundaries for segmentation task is a tedious and costly process, and most large-scale benchmark datasets such as *PubLayNet* (Zhong et al., 2019) and *DocLayNet* (Pfitzmann et al., 2022) only provide rectangular boundaries for the document entities. Such coarse annotations fail to capture the complex shapes and layouts of document objects (Harley et al., 2015; Clausner et al., 2019), limiting the generalization ability of the segmentation models. Therefore, it is essential to explore unsupervised or self-supervised segmentation approaches that can leverage unlabeled data.

The self-supervised methods for natural images rely on contrastive learning to distinguish different classes of objects (Chen et al., 2020). However, this approach is not suitable for document images as they contain multiple objects of various classes in a complex layout. Moreover, feature matching methods that use global and local views without object localisation lose information and fail to capture the document structure and content. Recent works (Li et al., 2021a; Huang et al., 2022) have used self-supervision for pre-training the encoder of document segmentation models, by combining mask language and image modeling to learn generic textual and visual features. However, these methods require supervised fine-tuning with annotated data for the segmentation task.

A vision-based self-supervised learning framework for DLA can address the limitations of existing methods, provided that the visual features are learned from reliable pseudo layout masks. However, the existing framework of *SelfDocSeg* (Maity et al., 2023) generates pseudo layout masks by applying morphological operations such as erosion, which are sensitive to the variations in the document characteristics. For example, the gray level intensity of text and image objects, the kernel size and shape for erosion, and the threshold value for binarization can affect the quality and consistency of the pseudo layout masks. Therefore, a fully automated *SelfDocSeg* segmentation pipeline may produce inaccurate or noisy masks during the training stage (Fig. 5). To overcome this issue, *SelfDocSeg* uses self-supervision as a pre-training step and fine-tunes the model on a specific dataset in supervised setting, but this compromises the generalization capability of the model across different document types.

In this paper, we present a self-supervised framework for document segmentation that does not require human annotations. It has two components: an unsupervised method to generate polygonal masks (isothetic covers) that approximate document objects, such as paragraphs, images, tables, etc, and a learning-based method to refine them using an encoder-decoder network. Our unsupervised method improves on Biswas et al. (2010)'s grid-based technique by applying local and global thresholding to capture intensity variations and reduce binarization effects. We also propose a polygon merging algorithm that integrates both thresholding results and removes redundant or overlapping polygons, while preserving the structure, alignment, and isothetic properties of the document objects. It should be noted that the output quality of isothetic cover based approaches is influenced by the type of document, quality of binarization technique used, noise and artifacts present in the document, etc. To counter the above drawbacks, we introduce a learning based approach which leverages these pseudo masks as reference. The network learns to correct the inconsistencies and generalize to different document types, using a large-scale dataset of diverse document images. Overall, our main contributions are as follows:

- We present AutoDocSegmenter, a self-supervised framework for end-to-end document-image segmentation that leverages isothetic covers as pseudo masks. AutoDocSegmenter can adopt any encoder-decoder segmentation model and we show its versatility and effectiveness with transformer-based and *ConvNet*-based encoders.

- We evaluate AutoDocSegmenter on *PRImA* (Antonacopoulos et al., 2009), *DocLayNet*, *PubLayNet*, and $M^6$-*Doc* (Cheng et al., 2023) datasets and show that it outperforms the existing baselines in both within domain and cross domain settings. Overall, we observe that AutoDocSegmenter is able to generalize on complex layout images which are not observed during the training stage.

Figure 1 shows sample segmentation results from *SelfDocSeg* (Maity et al., 2023), *SwinDocSegmenter* (Banerjee et al., 2023) and our AutoDocSegmenter models. Since the pre-trained *SelfDocSeg* model was unavailable, we followed train a MobileNetV3-small backbone (MNV3sm) using *SelfDocSeg*'s pseudo mask generation method (Maity et al., 2023). We observe that our method can segment various types of document images such as magazine and scientific report images. In contrast, *SwinDocSegmenter* and *SelfDocSeg* struggle to handle the diversity and complexity in document layouts.

The rest of the paper is as follows. Section 2 discusses related works. We introduce our self-supervised framework, AutoDocSegmenter, and discuss its components in Section 3. In Section 4, we present a comprehensive analysis of the experiments conducted to validate our approach, along with a detailed discussion of the observed results. Section 5 concludes the paper.

## 2 Related Work

In this section we discuss classical as well as recent deep learning based DLA approaches.

### 2.1 Heuristic Rule-based Methods

Rule-based methods are typically classical approaches, which perform document segmentation by using pixel-level information. Based on the sequence in which segmentation is performed, these can be categorized as top-down, bottom-up, or hybrid. In top-down methods (Kise et al., 1998; Journet et al., 2005), the entire document is successively split into various components at each step and further into definite regions depicting similar entities. Bottom-up techniques (Saabni & El-Sana, 2011; Asi et al., 2015) are usually more effective than top-down methods. They begin by considering every pixel as a separate cluster and at successive iterations, similar pixels are grouped together until homogeneous regions are formed. Hybrid technique (Tran et al., 2015) combine both the above approaches to obtain fast and generalized segmentation methods.

### 2.2 Using Convolutional Architecture

Over the last decade, *ConvNet* have been widely used for DLA and in particular for document segmentation. Early works for page segmentation (He et al., 2017a; Li et al., 2020b) explored CNNs for extracting effective

Table 1: Comparing the training characteristics of the state-of-the-art techniques with our proposed approach.

| Method | Text features | Image features | Pseudo masks | Self-supervised pre-training | Supervised fine-tuning | Self-supervised training |
|---|---|---|---|---|---|---|
| Layout Parser (Shen et al., 2021) | ✓ | ✓ | × | × | ✓ | × |
| DocSegTr (Biswas et al., 2022) | × | ✓ | × | × | ✓ | × |
| LayoutLMv3 (Huang et al., 2022) | ✓ | ✓ | × | ✓ | ✓ | × |
| UDoc (Gu et al., 2021) | ✓ | ✓ | × | ✓ | ✓ | × |
| DiT$_{BASE}$ (Li et al., 2022) | × | ✓ | × | ✓ | ✓ | × |
| MaskRCNN (He et al., 2017b) | × | ✓ | × | × | ✓ | × |
| FasterRCNN (Ren et al., 2015) | × | ✓ | × | × | ✓ | × |
| SwinDocSegmenter (Banerjee et al., 2023) | × | ✓ | × | × | ✓ | × |
| BYOL (Grill et al., 2020) | × | ✓ | × | ✓ | ✓ | × |
| SelfDocSeg (Maity et al., 2023) | × | ✓ | ✓ | ✓ | ✓ | × |
| **AutoDocSegmenter** (ours) | × | ✓ | ✓ | × | × | ✓ |

visual features. Several CNN based segmentation approaches have focused on specific document types such as historical documents (Chen et al., 2015; 2017; Oliveira et al., 2018), newspaper articles (Almutairi & Almashan, 2019), and scientific publications (Biswas et al., 2021; Yang & Hsu, 2021). Researchers have also developed methods on recognizing and segmenting specific objects. For instance, *DeepDeSRT* (Schreiber et al., 2017) and *CascadeTabNet* (Prasad et al., 2020) aimed at detecting and segmenting tables. Lin et al. (Lin et al., 2021) developed a character (text region) detection technique based on *RetinaNet* (Lin et al., 2017) and transfer learning. Saha et al. (2019) used transfer learning on Faster-RCNN backbone to segment graphical objects in documents. To account for the domain shift in the documents present in training and test datasets, Li et al. (Li et al., 2020a) proposed cross-domain document object detection. *LayoutParser* (Shen et al., 2021) is a repository of pre-trained CNN models for layout detection, character recognition, and various other document processing tasks.

Recently, Maity et al. (2023) developed *SelfDocSeg* which performs self-supervised pre-training for document-image segmentation by combining Bootstrap Your Own Latent (BYOL) method (Grill et al., 2020) and focal loss on the pseudo masks. The pseudo masks are generated using morphological operations such as erosion. However, erosion-based masks potentially have drawbacks such as: (i) they may not not cover the document objects entirely; (ii) they may distort the structure and shape of the document objects; and importantly (iii) they are sensitive to the resolution of the image and the size and shape of the erosion kernel. To overcome such issues, *SelfDocSeg* finally fine-tune the network with labeled datasets.

## 2.3 Using Transformer Architecture

Transformer based models gained popularity in computer vision after the success of Vision Transformers (ViT) (Dosovitskiy et al., 2021). Li et al. (Li et al., 2022) proposed self-supervised document image transformers *DiT* and showed its effectiveness in various downstream tasks like document classification, layout analysis, table detection, and text detection (OCR). However, the pre-training for *DiT* is done on large-scale unlabelled document images and is not applicable to small scale magazine datasets (e.g., *PRImA*) with dataset-specific attributes (Banerjee et al., 2023). *StrucTexT* (Li et al., 2021b) employed multi-modal transformers for understanding structured text. Although *StrucTexT* obtains good performance at both segment and token levels, it gets confused when the textual contents are semantically related and closely placed. Yang & Hsu (2022) employ OCR based text extraction to perform segmentation on *PubLayNet* dataset. *LayoutLM* (Xu et al., 2020) and *LayoutLMv3* (Huang et al., 2022) perform joint learning of layout, visual, and text features for visual document understanding tasks.

Biswas et al. (Biswas et al., 2022) proposed *DocSegTr* using *ResNet*-FPN backbone on transformer. Since *DocSegTr* employs self attention mechanism, it is able to achieve faster convergence on small-scale datasets and obtain competitive performance. *SwinDocSegmenter* (Banerjee et al., 2023) is state-of-the-art instance segmentation method on complex layout document images. It employs a supervised SwinTransformer feature extractor backbone. This is integrated with a transformer-based encoder-decoder architecture which

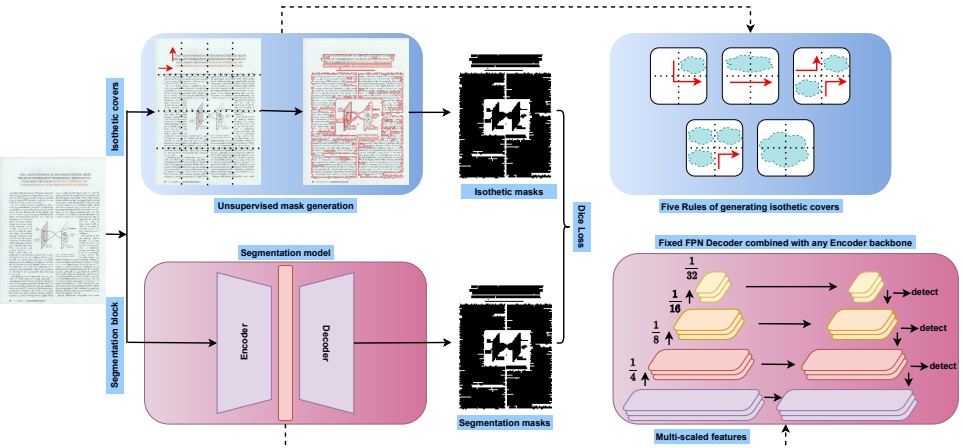

Figure 2: Schematic diagram of our framework, AutoDocSegmenter. The isothetic covers are generated using five types of rules Section 3.1 and the isothetic masks extracted are used as reference to train an encoder-decoder model. Different encoder backbones (e.g., MNV3sm, MiT-B0, etc.) may be employed. We fix the decoder network to feature pyramid network (FPN).

fine-tunes on the downsampled features from the backbone using the annotations in a supervised learning paradigm. Some approaches (Li et al., 2021a; Kim et al., 2022) use cross-modality encoder during self-supervised pre-training phase to recognise the types of documents and segment entities. Recent techniques (Appalaraju et al., 2021; Kim et al., 2022; Gu et al., 2021; 2022) have also focused on general purpose features which are generated using unified pre-training on multiple downstream tasks. However, such methods have inherent biases towards classes with more number of samples and fail to perform during domain shift.

### 2.4 Distinguishing AutoDocSegmenter with Existing Methods

The existing (partially) self-supervised methods (Huang et al., 2022; Maity et al., 2023; Banerjee et al., 2023) mainly employ self-supervision for pre-training and subsequently perform a supervised fine-tuning step for dataset specific segmentation. The supervised dataset is typically annotated using rectangular boundaries which vaguely distinguishes different objects present in the image. However, for a detailed document analysis, it is essential to capture the structure of these entities. On the other hand, our fully self-supervised approach, AutoDocSegmenter, is able to annotate as well as segment the document images, using polygonal boundaries to outline different objects present in the document. Empirically, AutoDocSegmenter generalize well to various document layouts without additional fine-tuning. We enlist the distinctions of our method compared to the existing deep-learning based document segmentation approaches in Table 1. We note that all the methods in Table 1 except our AutoDocSegmenter have a supervised fine-tuning step.

## 3 Proposed approach: AutoDocSegmenter

In this section, we discuss our self-supervised framework for document segmentation. Fig. 2 illustrates the main components and workflow of AutoDocSegmenter. In Section 3.1, we discuss unsupervised pseudo masks generation mechanism. The pseudo masks are subsequently used to train the segmentation model, as detailed in Section 3.2.

### 3.1 Robust isothetic covers using local and global information

Isothetic covers are axis-aligned polygons that encloses the objects in binary images. They capture object structure and geometry with low cost. Variation in document type and binarization quality may render the isothetic covers noisy. In particular, binarization is challenging because a single/global (binarization) threshold may not preserve all document objects, while multiple local thresholds may introduce noise arti-

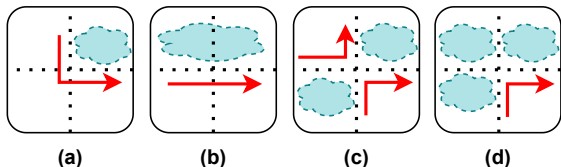

Figure 3: The directions of traversal depending on the occupancy of the four grids: (a) type 1 vertex with exactly one grid occupied, (b) type 2 vertex where adjacent grid are occupied, (c) type 3 vertex with diagonal vertex occupied, and (d) type 4 vertex with exactly three grids occupied.

facts. Hence, we propose a novel isothetic cover algorithm using both local and global binarized images. In the following, we first detail the vanilla isothetic cover algorithm (Biswas et al., 2010) and then present our improved isothetic cover algorithm.

**Isothetic covers:** To form polygons that enclose the objects with minimal area, an input image is divided into grids using fixed-distance horizontal and vertical lines between them. The grids are traversed in a row-major order, and the vertices of the polygons are determined by the occupancy of the four grids surrounding each grid point, which indicates whether or not objects are present within them. The method classifies the grid points as vertices of different types and angles using six criteria (Biswas et al., 2010):

1. A grid point is not a vertex and is skipped if none of the four grids are occupied.

2. A grid point is a vertex of type 1 with a 90° angle and a value of $t = 1$ if exactly one of the four grids is occupied. The method turns to the occupied grid and continues along the same row or column (Fig. 3 (a)).

3. A grid point is a vertex of type 0 with a 180° angle and a value of $t = 0$ if two adjacent grids are occupied. The method follows the edge between the occupied grids and moves to the next row or column (Fig. 3 (b)).

4. A grid point is a vertex of type -1 with a 270° angle and a value of $t = -1$ if two diagonal grids are occupied. The method turns to the unoccupied grid and advances along the same row or column (Fig. 3 (c)).

5. A grid point is also a vertex of type -1 with a 270° angle and a value of $t = -1$ if three of the four grids are occupied. The method turns to the unoccupied grid and proceeds along the same row or column (Fig. 3 (d)).

6. A grid point is not a vertex and is skipped if all four grids are occupied.

The detailed algorithm in discussed in the Appendix A.2. The isothetic covers are generated without any backtracking and in linear time with respect to the perimeter of the polygon. Pseudo masks are generated by filling these polygons. We refer to this stage of training pipeline with the conventional isothetic covers (Biswas et al., 2010) as AutoDocSegmenter-I.

**Binarization and Merging polygons:** Binarization facilitates the extraction of isothetic polygonal covers from document images, by enabling a clear distinction between foreground and background regions. However, the quality of binarization depends on the choice of the threshold value, which can affect the preservation of information or the exclusion of noise. Different document objects may have different gray levels, making it challenging to find a single global threshold that can separate them from the background. On the other hand, applying multiple local thresholds on small image patches may introduce noise artifacts such as bleed-through, speckles, or uneven illumination. We introduce a hybrid method for obtaining the isothetic covers from a document image, which combines the advantages of global and local thresholding techniques. Importantly, our method maintains the shape and orientation of the document elements, and produces isothetic polygons that are aligned with the axes and independent of the resolution. This facilitates effective polygon merging with minimal loss or distortion of information.

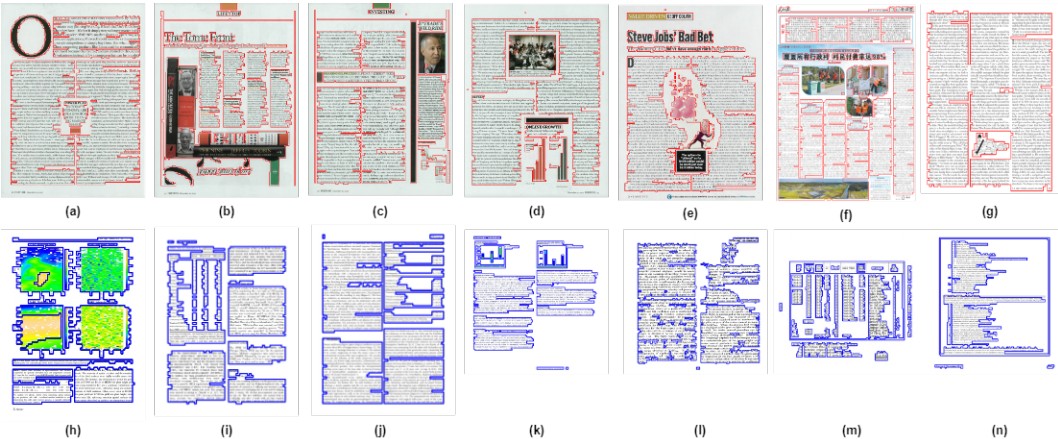

Figure 4: Annotating documents using isothetic covers. (a)-(e) Results from *PRImA* dataset, (f)-(g) Results from $M^6$-*Doc* dataset, (h)-(j) Results from *PubLayNet* dataset, and (k)-(n) Results from *DocLayNet*. The algorithm is able to annotate complex layouts of in-line images, tables, circuits and graphs.

For each image ($I$), we generate two types of binarized images: 1) a binary image obtained from global thresholding ($B_g(I)$), and 2) a binary image obtained from local thresholding ($B_l(I)$). For local thresholding, we divide the image into non-overlapping sliding windows of size $w \times w$ and apply binarization technique on each window independently, where $w$ is computed as 1/5th of the minimum dimension of the image. We then apply the isothetic covers algorithm on both the binary images to obtain two sets of isothetic polygons that approximate the document objects. To merge the two sets of polygons, we propose a novel Algorithm 1 that aims to preserve the information from both the global and local binarization. The selected polygons are then considered as the final set of isothetic polygons. We then use the region filled isothetic polygons as pseudo-segmentation masks to train our segmentation model. We refer to this stage of the training pipeline with our modified isothetic covers as AutoDocSegmenter-U. Thus, while AutoDocSegmenter-I generates pseudo masks using the conventional isothetic covers (Biswas et al., 2010), AutoDocSegmenter-U generates pseudo masks using the proposed (Algorithm 1).

Fig. 4 shows the polygonal masks for various document objects obtained using AutoDocSegmenter-U on *PRImA*, $M^6$-*Doc*, *PubLayNet*, and *DocLayNet* datasets. We observe that the obtained (unsupervised) polygonal masks approximate the shapes of the enclosed objects. Thus, they provide a coarse but reasonable supervision for the (next) model to learn the boundaries of the document objects.

## 3.2 Segmentation Model

We adopt a feature pyramid network (FPN) decoder based segmentation architecture for our document segmentation task. FPN is a general-purpose decoder that can work with various encoder backbones and

---

**Algorithm 1** Merging Polygons

---

$S_g = \{\text{x} \mid \text{x is a point in Polygon } P_g \text{ of } B_g(I)\}$
$S_l = \{\text{ x' } \mid \text{ x' is a point in Polygon } P_l \text{ of } B_l(I)\}$
$P_f = \text{Final set of merged polygons}$
**if** $S_g == S_l$ **then**
    $P_f \leftarrow P_g \parallel P_l$
**else if** $S_l \subseteq S_g$   &   $|S_l \cap S_g| \geq 0.5|S_g|$ **then**
    $P_f \leftarrow P_g$
**else if**   $S_g \subseteq S_l$   &   $|S_l \cap S_g| \geq 0.5|S_l|$ **then**
    $P_f \leftarrow P_l$
**end if**

---

produce multi-scale feature maps from a single input image. FPN is suitable for our task as it can capture both fine and coarse details of the document layout.

**Encoder block:** Our self-supervised method does not rely on any specific encoder architecture or pre-training scheme, as it only uses conventional learning of isothetic masks. Therefore, we can integrate our method with different types of encoder backbones, ranging from lightweight to heavyweight models, such as *EfficientNets* (Tan & Le, 2019), *MobileNetV2* (Sandler et al., 2018), *MobileNetV3* (Howard et al., 2019), *Mix Transformer* (Xie et al., 2021), *ResNeSt* (Zhang et al., 2022), *MobileOne* (Vasu et al., 2023), *ResNeXt* (Xie et al., 2017), or *RegNet* (Xu et al., 2022). These encoder backbones extract feature maps at different levels of resolution and semantic richness from the input image.

**Decoder block:** Document segmentation is a challenging task that requires capturing both fine-grain and coarse features of diverse real-world documents. A single architectural choice for the encoder-decoder network may compromise the precision of the segmentation masks. To overcome this limitation, we adopt the Feature Pyramid Network (FPN) decoder, which produces multi-scale feature maps by fusing low-level and high-level features from the encoder backbone. The FPN decoder leverages the spatial resolution of the low-level features and the semantic information of the high-level features to enhance the accuracy and robustness of the document segmentation. In particular, the FPN decoder fuses the multi-scale features from the encoder backbone through a top-down pathway and lateral connections. The top-down pathway upsamples the highest level encoder feature map progressively by nearest neighbour interpolation and adds it element-wise to the corresponding encoder feature map, which is first reduced to the same number of channels by a $1 \times 1$ convolution. This fusion process is repeated for all encoder levels, creating a set of merged feature maps. To reduce the anti-aliasing effect of upsampling at each level a $3 \times 3$ convolution is applied. Finally, the refined feature maps are passed through a common classification layer to obtain the final segmentation mask.

**Loss function:** Our model aims to segment document images using polygonal masks where the reference annotation is the isothetic covers generated in an unsupervised manner. We employ the Dice loss between the generated and reference masks. Let $y$ represents the pseudo annotations generated using isothetic covers and $\bar{p}$ represents the predicted segmentation generated by the model. Then, the Dice loss is computed as

$$L(y, \bar{p}) = 1 - \frac{2y\bar{p} + 1}{y + \bar{p} + 1}. \tag{1}$$

The formula $\frac{2y\bar{p}+1}{y+\bar{p}+1}$ calculates a value between 0 and 1, with 0 indicating no similarity between the two segmentation, and 1 indicating complete similarity. Thus, the loss become 0 when the masks are completely similar. The model is trained using Adam optimizer.

## 4 Experiments

We evaluate the proposed AutoDocSegmenter approach on different benchmarks datasets by (a) comparing it against state-of-the-art document segmentation methods (Section 4.2) and (b) performing ablation studies with varying backbone feature extractors (Section 4.3.1), polygon merging thresholds (Section 4.3.2), binarization techniques (Section 4.3.3), grid size of isothetic covers (Section 4.3.4), image sizes (Section 4.3.5), and document object types (Section 4.3.6). We begin by detailing our experimental setup.

### 4.1 Experimental Details

**Datasets**: We evaluate our method on four popular document segmentation datasets: *PRImA* (Antona-copoulos et al., 2009), *DocLayNet* (Pfitzmann et al., 2022), *PubLayNet* (Zhong et al., 2019), and $M^6$-*Doc* (Cheng et al., 2023). *PRImA* has 382 and 96 images for training and testing, respectively, with polygonal annotations for each entity. *DocLayNet* and *PubLayNet* have rectangular annotations and contain 69 375/6489 and 335 703/11 245 images for training/testing, respectively. $M^6$-*Doc* is a recent dataset with only a test set of 2724 images. Please refer to Appendix A.1 for more details.

**Evaluation Metric:** We use the mean average precision (mAP) metric to compare segmentation models, which averages the intersection over union (IoU) scores of predicted and groundtruth masks across documents

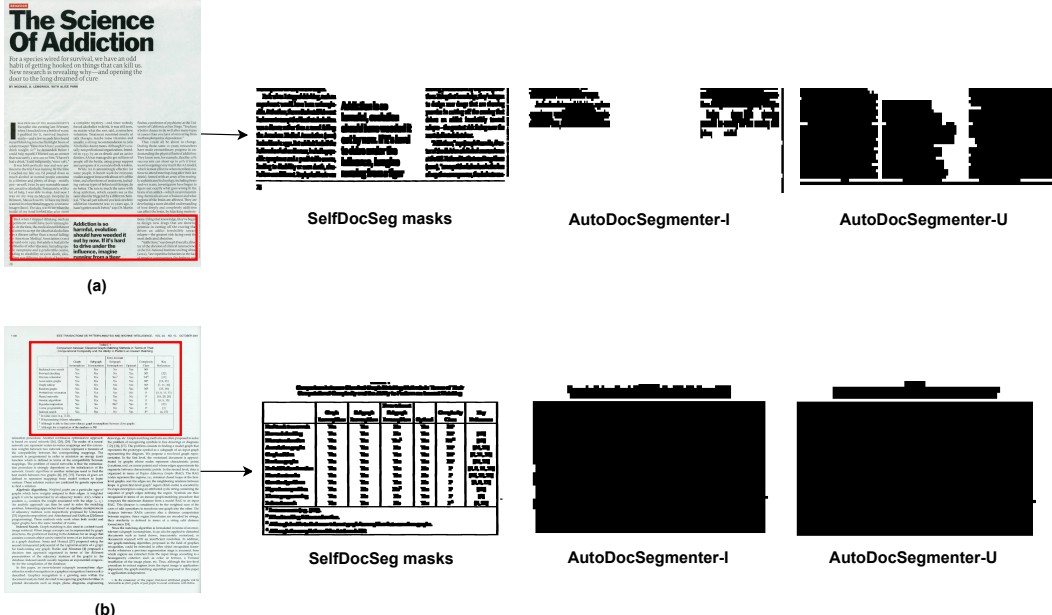

Figure 5: Comparison of pseudo masks generated by SelfDocSeg Maity et al. (2023) and AutoDocSegmenter-I with the modified AutoDocSegmenter-U. (a) This is an example for a document image where the AutoDocSegmenter-I fails to capture texts at the bottom written in bold while the AutoDocSegmenter-U covers these regions. (b) SelfDocSeg does not mask the table region in the image unlike AutoDocSegmenter-U which precisely covers it.

and IoU thresholds from 0.5 to 0.95 in steps of 0.05. This follows the Microsoft COCO benchmark protocol (Banerjee et al., 2023).

**AutoDocSegmenter training and evaluation:** As discussed in Section 3.1, the first stage of our pipeline generate pseudo masks for each object without using any supervision. For both isothetic covers as well as the proposed modified isothetic covers (Algorithm 1), we employ Otsu's thresholding technique (Otsu, 1979) for global and local thresholding, as it can automatically find the optimal threshold that minimizes the intra-class variance of the pixel intensities. The model is trained with the Adam optimizer and a learning rate of 0.001 for 50 epochs, and the learning rate is lowered by a factor of 10 every 10 epochs. All document images are resized to 256 pixels. We do not utilize any annotations from the datasets during training. We assess the model's segmentation quality by comparing its results with the groundtruth labels from the datasets.

## 4.2 Experimental Analysis

In this section, we compare our AutoDocSegmenter approach with the existing approaches in terms of the pseudo-mask generation quality as well as the overall segmentation generalization performance.

### 4.2.1 Comparison of pseudo masks

To evaluate the effectiveness of our dual binarization and polygon merging algorithm, we first compare the masks generated by AutoDocSegmenter-U with the existing isothetic covers algorithm (AutoDocSegmenter-I) (Biswas et al., 2010) and erosion-based SelfDocSeg Maity et al. (2023) masks in Fig. 5.

Fig. 5(a) shows that AutoDocSegmenter-I fails to segment the bold text at the bottom of the image, as it relies on gray level intensities that vary across the document. The erosion-based masks produced by SelfDocSeg (Maity et al., 2023) create a binary version of the original image, where text elements are merged together. This does not provide useful segmentation masks for training models, as it loses the distinction between different text regions. While the erosion window size of SelfDocSeg can be adjusted to improve the

Table 2: Comparing the mean average precision (mAP) of AutoDocSegmenter-U and SelfDocSeg masks on *PRImA*, *DocLayNet* and *PubLayNet* datasets.

| Masks | *PRImA* | *DocLayNet* | *PubLayNet* |
|---|---|---|---|
| SelfDocSeg mask (Maity et al., 2023) | 66.67 | 77.0 | 79.90 |
| AutoDocSegmenter-U | 82.20 | 82.90 | 84.40 |

Table 3: Comparing the mean average precision (mAP) when each model is trained on *PubLayNet*, and evaluated on *PRImA*, *DocLayNet* and $M^6$-*Doc* datasets.

| Method | PRImA | *DocLayNet* | $M^6$-*Doc* |
|---|---|---|---|
| LayoutLMv3 (Huang et al., 2022) | 4.35 | 2.30 | 5.30 |
| SwinDocSegmenter (Banerjee et al., 2023) | 43.80 | 10.00 | 33.40 |
| AutoDocSegmenter-I-MiT-B0 (ours) | 89.57 | 83.21 | 75.22 |
| AutoDocSegmenter-I-MNV3sm (ours) | 85.46 | 83.43 | 74.26 |
| AutoDocSegmenter-U-MiT-B0 (ours) | **89.65** | **83.80** | **75.56** |
| AutoDocSegmenter-U-MNV3sm (ours) | 87.76 | 83.66 | 75.24 |

text segmentation, it does not affect the segmentation of other objects. In contrast, AutoDocSegmenter-U segments the document by tracing the boundaries of the objects, resulting in well-defined masks for each text region. The difference in the masks is more evident in figures, tables and graphs, where AutoDocSegmenter-U groups each object as a single unit, while SelfDocSeg and AutoDocSegmenter-I split them into multiple parts, as seen in Fig. 5(b).

We also evaluate the pseudo masks from SelfDocSeg (Maity et al., 2023) and AutoDocSegmenter-U methods with respect to the given annotations in Table 2. We observe that AutoDocSegmenter-U outperforms SelfDoc-Seg by extracting higher quality pseudo-masks on all the three datasets. Overall, our AutoDocSegmenter-U preserves the structure and alignment of the document objects while ensuring that they remain axis-parallel and resolution-independent. This enables precise capturing of the structure of the objects with minimal information loss or distortion. On the other hand, morphological operation-based SelfDocSeg Maity et al. (2023) does not capture the shape of the object precisely and distorts with varying intensities.

### 4.2.2 Comparison against existing approaches

We next compare AutoDocSegmenter against recent document segmentation approaches in two settings: (1) methods are trained on a large unannotated corpus (*PubLayNet*) and evaluated on unseen datasets (*PRImA*, *DocLayNet*, and $M^6$-*Doc*), and (2) training and test sets belong to the same dataset. In this section, we report results of AutoDocSegmenter with lightweight encoders (MiT-B0 and MobileNetV3-small) which offer significant advantages in terms of parameter and computational efficiency. These are important prerequisites for mobile applications (e.g., Microsoft's M365 and Office Lens apps, Adobe Scan app, etc.) that demand fast and reliable document layout analysis. Thus, such variants of AutoDocSegmenter enhance its applicability in real-world scenarios. In Section 4.3, we also evaluate the performance of AutoDocSegmenter with various feature extractors.

**Setting 1.** We evaluate the generalization ability of our method and the baselines in the scenario where the training and test sets come from different sources. This is a realistic situation where one has access to a large amount of unlabeled images for training and wants to apply the segmentation model to a specific dataset (e.g., new user data) that may have different characteristics. The baselines we compare with are *LayoutLMv3* (Huang et al., 2022) and *SwinDocSegmenter* (Banerjee et al., 2023), whose pre-trained models (on *PubLayNet* dataset) are publicly available. Table 3 shows the performance of different methods when they are trained on *PubLayNet* and tested on *PRImA*, *DocLayNet* or $M^6$-*Doc*.

We observe in Table 3 that our AutoDocSegmenter approach achieves the best results on all the three test datasets, outperforming both state-of-the-art baselines by a large margin. This indicates that our

Table 4: Comparing the mean average precision (mAP) of our proposed method with state-of-the-art techniques on *PRImA*, *PubLayNet* and *DocLayNet* datasets. **M** denotes number of parameters in millions.

| Method | M | *PRImA* | *DocLayNet* | *PubLayNet* |
|---|---|---|---|---|
| Layout Parser (Shen et al., 2021) | - | 64.70 | - | 86.70 |
| DocSegTr (Biswas et al., 2022) | - | 42.50 | - | 90.40 |
| LayoutLMv3 (Huang et al., 2022) | 133 | 40.30 | - | **95.10** |
| UDoc (Gu et al., 2021) | 272 | - | - | 93.90 |
| DiT$_{BASE}$ (Li et al., 2022) | 87 | - | - | 93.50 |
| MaskRCNN (He et al., 2017b) | 63.7 | - | 73.50 | 91.00 |
| FasterRCNN (Ren et al., 2015) | 19 | - | 73.40 | 90.20 |
| SwinDocSegmenter (Banerjee et al., 2023) | 223 | 54.39 | 76.85 | 93.72 |
| BYOL (Grill et al., 2020) | 94 | 28.70 | 63.50 | 79.00 |
| SelfDocSeg (Maity et al., 2023) | - | 52.10 | 74.30 | 89.20 |
| AutoDocSegmenter-I-MiT-B0 (ours) | 5 | 79.12 | **87.20** | 84.50 |
| AutoDocSegmenter-I-MNV3sm (ours) | 4.1 | 80.72 | 87.14 | 85.10 |
| AutoDocSegmenter-U-MiT-B0 (ours) | 5 | 80.51 | 82.50 | 85.20 |
| AutoDocSegmenter-U-MNV3sm (ours) | 4.1 | **81.47** | 85.97 | 85.20 |

approach generalizes well to different types of documents and captures the layout information effectively. Both LayoutLMv3 and SwinDocSegmenter are large-scale transformer-based models. We note that both the above baselines struggle to adapt to the diverse domains and styles of the documents, and may suffer from overfitting to the *PubLayNet* dataset.

We also note from Table 3 that AutoDocSegmenter-MiT-B0 has a slight edge over AutoDocSegmenter-MNV3sm. MiT-B0 is based on the transformer architecture, which can capture the long-range dependencies and the spatial relations in the documents, while MNV3sm is based on the mobile network architecture, which can reduce the computational cost and the runtime memory usage.

**Setting 2.** We next evaluate the models when the training and test sets belong to the same dataset in Table 4. In addition to *LayoutLMv3* and *SwinDocSegmenter*, we also report the results obtained by *Layout Parser* (Shen et al., 2021), *SelfDocSeg* (Maity et al., 2023), *DocSegTr* (Biswas et al., 2022), *MaskRCNN* (He et al., 2017b), and *FasterRCNN* (Ren et al., 2015). For *LayoutLMv3* and *SwinDocSegmenter*, we obtain the results using the pre-trained weights provided by the authors while for other baselines, we report the results as published in (Shen et al., 2021; Biswas et al., 2022; Zhong et al., 2019). Overall, AutoDocSegmenter demonstrates superior performance across datasets. In particular, we make the following observations:

- *PRImA* dataset contains complex and diverse document layouts with non-rectangular annotation boundaries. The results in Table 4 show that AutoDocSegmenter is able to handle the variations and challenges of different document shapes and structures.

- *DocLayNet* dataset consists of document images with rectangular annotation boundaries and various document structures. AutoDocSegmenter outperforms the baselines on this dataset, indicating its robustness and adaptability to different layout styles.

- *PubLayNet* is a large dataset, consisting of research document images with non-overlapping rectangular annotation boundaries and relatively simple and regular layouts. LayoutLM3 is the best performing method on this dataset.

In the model training phase with unsupervised annotations, we observe two kinds of outcomes: (a) the model mimics some of the annotation errors, such as careful outlining of in-line images, in the segmentation masks, and (b) the model deviates from some of the annotation errors, and thus discards some of the wrongly labeled areas. These outcomes indicate both the advantages and disadvantages of using an unsupervised annotation process. However, as shown in Table 3, AutoDocSegmenter can generalize effectively to various unseen

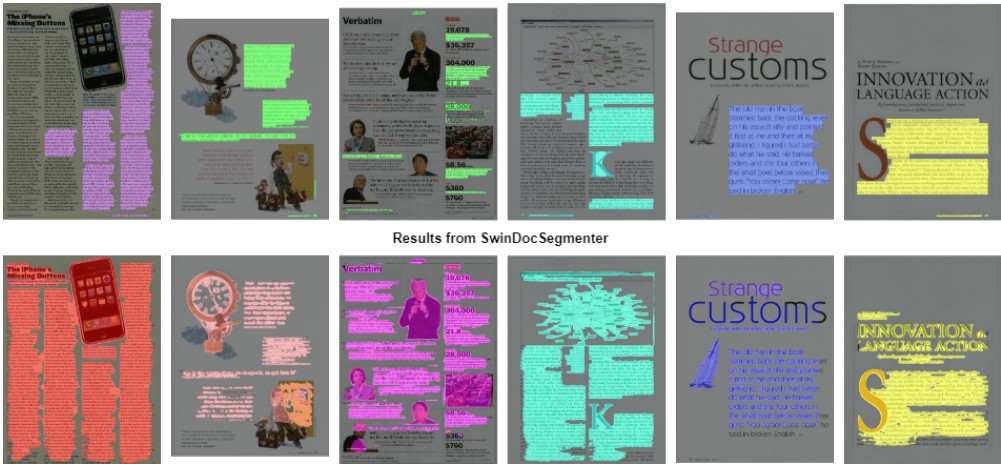

Figure 6: Comparing the segmentation results from *PRImA* datasets using AutoDocSegmenter-MiT-B0 model with *SwinDocSegmenter*. Our algorithm captures the layout of the objects more precisely compared to *SwinDocSegmenter* (Banerjee et al., 2023).

datasets, which lowers its dependence on data-specific hyperparameters. Hence, we see that AutoDocSegmenter achieves better performance on the *PRImA* dataset when self-supervised on the *PubLayNet* dataset (Table 3) than when self-supervised on the *PRImA* dataset (Table 4).

Fig. 6 shows some examples of the segmentation results from the *PRImA* dataset using our AutoDocSegmenter-U-MiT-B0 model. We observed that our proposed approach can capture complex structures, such as tables, figures, captions, and in-line images, which are often missed by SwinDocSegmenter. This shows that AutoDocSegmenter can handle various document layouts and content types, and produce accurate segmentation masks.

### 4.3 Ablation studies

To evaluate the impact of our training parameters, we now discuss ablation studies performed with the proposed AutoDocSegmenter framework under different settings.

#### 4.3.1 Performance on various backbone feature extractors

We begin by evaluating the performance of AutoDocSegmenter-U using different CNN and transformer based backbone feature extractors (combined with the FPN decoder). We train the models using image size of 256. Table 5 compares the performance of AutoDocSegmenter-U on various backbone feature extractors on two datasets: *PRImA* and *DocLayNet*. In addition to mAP metric, we also report intersection over union (IoU) scores. We observe from Table 5 that:

- The transformer-based MiT-B4 performs worse than the much lighter MiT-B0 backbone or the CNN-based backbones. This may indicate that it requires more data and fine-tuning to achieve comparable results.

- Among the CNN-based backbones, AutoDocSegmenter-U-EfficientNet variants are among the best methods on both datasets. However, these backbones also have the largest number of parameters, ranging from 5.3M to 30M, which may limit their applicability on resource-constrained scenarios.

- AutoDocSegmenter-U-MobileNet variants, with MobileNetV2 and MobileNetV3 sm backbones, show competitive performance on both datasets. The number of parameters in these backbones range from 2.5M to 3.4M, making them efficient and lightweight for mobile scenarios.

Table 5: Comparison of the performance of AutoDocSegmenter-U on various backbone feature extractors on *PRImA* and *DocLayNet* datasets. **M** denotes number of parameters in millions.

| Backbone | M | *PRImA* | | *DocLayNet* | |
|---|---|---|---|---|---|
| | | IoU | mAP | IoU | mAP |
| EfficientNet-B0 (Tan & Le, 2019) | 5.3 | 0.65 | 84.11 | 0.83 | 97.56 |
| EfficientNet-B3 (Tan & Le, 2019) | 12 | 0.65 | **86.20** | 0.84 | 98.80 |
| EfficientNet-B5 (Tan & Le, 2019) | 30 | **0.69** | 82.42 | 0.87 | **99.20** |
| MobileNetV2 (Sandler et al., 2018) | 3.4 | 0.65 | 80.01 | 0.86 | 96.37 |
| MobileNetV3 sm (Howard et al., 2019) | 2.5 | 0.67 | 81.47 | 0.84 | 96.30 |
| MobileOne-S0 (Vasu et al., 2023) | 2.1 | 0.42 | 42.70 | 0.70 | 70.20 |
| RegNetX-200 (Xu et al., 2022) | 2.7 | 0.42 | 42.70 | 0.70 | 70.20 |
| ResNeSt-14 (Zhang et al., 2022) | 8 | 0.67 | 84.08 | 0.84 | 96.87 |
| ResNeSt-26 (Zhang et al., 2022) | 15 | 0.63 | 75.73 | 0.83 | 97.10 |
| ResNeSt-50 (Zhang et al., 2022) | 25 | 0.64 | 79.26 | 0.88 | 98.20 |
| ResNeSt-101 (Zhang et al., 2022) | 46 | 0.62 | 82.83 | **0.89** | 98.62 |
| ResNeXt-50 (Xie et al., 2017) | 22 | 0.65 | 75.70 | **0.89** | 98.30 |
| MiT-B0 (Xie et al., 2021) | 3 | 0.67 | 80.51 | 0.81 | 93.64 |
| MiT-B4 (Xie et al., 2021) | 60 | 0.66 | 80.33 | 0.70 | 70.20 |

Table 6: Analysing the performance of AutoDocSegmenter-U with different overlapping threshold while merging on *PRImA* and *DocLayNet* datasets.

| Threshold (%) | *PRImA* | | *DocLayNet* | |
|---|---|---|---|---|
| | IoU | mAP | IoU | mAP |
| 25 | 0.71 | 77.60 | 0.78 | 85.00 |
| 50 | 0.74 | 82.20 | 0.79 | 82.90 |
| 75 | 0.67 | 72.70 | 0.79 | 85.20 |

- Other CNN-based backbones, such as MobileOne-S0, RegNetX-200, ResNeSt variants, and ResNeXt-50, have mixed results on both datasets.

In the following, we perform additional studies on AutoDocSegmenter-U-MiT-B0 and AutoDocSegmenter-U-MobileNetV3 small as they are among the lightest models with transformer-based and CNN-based backbones, respectively, and achieve reasonably good generalization performance.

### 4.3.2 Performance with different polygon merging thresholds

AutoDocSegmenter-U uses a polygon merging technique to mitigate the influence of binarization thresholds on the overall quality of pseudo masks. The polygons generated using local and global thresholding are merged if there is an overlap of more than a given threshold. Table 6 analyses the performance of AutoDocSegmenter-U using different overlapping threshold values on *PRImA* and *DocLayNet* datasets. We observe that the default threshold value of 50% is a robust choice across datasets. Setting it too low or too high may result in over- or under-segmentation, as observed in the *PRImA* dataset.

### 4.3.3 Performance with different binarization techniques

Document images often have regions with different gray levels, which pose a challenge for foreground-background separation using a single threshold value. For instance, documents with text and images, varying background colors, or noise effects such as shadows and low lighting, require different thresholds for different parts of the image to preserve the object shapes and details. A global threshold may fail to capture the contrast or brightness variations across the image Yan et al. (2005), while a local threshold based on a fixed window size may miss some objects or introduce artifacts due to the local background variation Wanas et al.

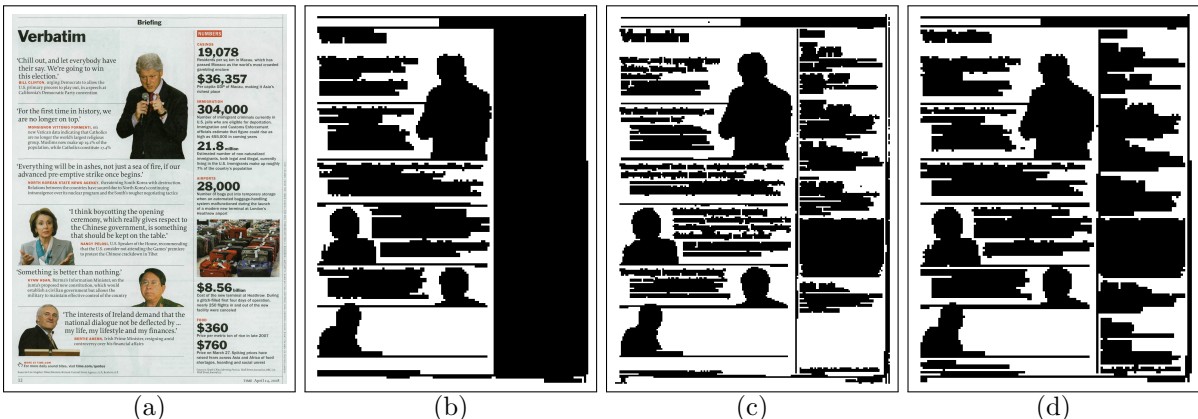

Figure 7: Pseudo masks generated for (a) the input image after binarisation using (b) Otsu's global thresholding, (c) Otsu's local thresholding and (d) Otsu's local+global thresholding. In (b), we observe that the global thresholding approach fail to differentiate between the foreground and the background in the right part of the document because of different grey-level of the background. In (c), we observe that the local thresholding approach unnecessarily captures very fined grained details, which would easily be influenced by noise. It also fails to capture objects near the bottom right of the document. In (d), the joint global and local approach combines the advantages of both the approaches and gives the best result.

(2006). To overcome these limitations, we explore an approach (Otsu's global + local thresholding) that adapts the threshold value to the global and local characteristics of the image. Fig. 7 shows the robust performance of this method over global or local thresholding alone on complex document images.

In Table 7, we compare the binarization performance of Otsu's global + local thresholding technique with the following baselines: (a) Global thresholding technique, where we set the threshold value to 147; (b) Otsu's global thresholding method (Otsu, 1979); (c) Adaptive mean thresholding, which is a local thresholding algorithm with window size fixed at 150; and (d) Otsu's local thresholding technique, where the threshold for every window is computed using Otsu's algorithm. In Table 7, we observe that Otsu's global + local thresholding performs competitively against the baselines while retaining the benefits of both local and global thresholding approaches. We note that our datasets contain limited number of complex images, which reduces the advantage of combining global and local thresholding techniques that can handle diverse document types. Therefore, we observe similar results from global thresholding and the combined method.

### 4.3.4 Varying grid size of isothetic covers

The quality of the isothetic masks is influenced by the grid size used for generating the isothetic covers. A suitable grid size should correspond to the scale and the variety of the document layout, preventing over-segmentation or under-segmentation of the layout elements. To measure the influence of the grid size on the segmentation performance, we calculate the intersection over union (IoU) and the mean average precision (mAP) of the isothetic masks with respect to the groundtruth masks.

Table 7: Analysing the performance of different binarization techniques in AutoDocSegmenter-U.

| Binarization method | PRImA | | DocLayNet | |
|---|---|---|---|---|
| | IoU | mAP | IoU | mAP |
| Global thresholding | 0.73 | 81.40 | 0.78 | 82.40 |
| Otsu's global thresholding | 0.74 | 82.40 | 0.79 | 86.00 |
| Adaptive mean thresholding | 0.72 | 79.80 | 0.77 | 82.00 |
| Otsu's local thresholding | 0.66 | 77.30 | 0.76 | 81.00 |
| Otsu's global + local thresholding | 0.74 | 82.20 | 0.79 | 82.90 |

Table 8: Analysing the performance of AutoDocSegmenter-U with different grid size for isothetic covers on *PRImA* dataset.

| Encoder | Grid size | IoU | mAP | AP@50 | AP@75 |
|---------|-----------|-----|-----|-------|-------|
| MiT-B0 | 10 | 0.42 | 42.70 | 42.70 | 42.70 |
| | 12 | 0.42 | 42.70 | 42.70 | 42.70 |
| | 15 | 0.43 | 42.70 | 42.70 | 42.70 |
| | 18 | **0.67** | **78.30** | **77.50** | **78.50** |
| MNV3sm | 10 | 0.55 | 56.71 | 55.80 | 57.50 |
| | 12 | 0.60 | 63.35 | 61.90 | 64.20 |
| | 15 | 0.64 | 70.94 | 69.50 | 71.80 |
| | 18 | **0.65** | **75.80** | **74.60** | **76.50** |

Table 9: Analysing the performance of AutoDocSegmenter with different image sizes on *PRImA* dataset.

| Encoder | Image size | IoU | mAP | AP@50 | AP@75 |
|---------|-----------|-----|-----|-------|-------|
| MiT-B0 | 1024 | 0.67 | **80.36** | **79.60** | **80.50** |
| | 512 | **0.69** | 79.53 | 78.80 | 79.70 |
| | 256 | 0.67 | 78.30 | 77.50 | 78.50 |
| MNV3sm | 1024 | **0.67** | **80.31** | **79.20** | **80.90** |
| | 512 | 0.66 | 77.53 | 76.20 | 78.10 |
| | 256 | 0.65 | 75.80 | 74.60 | 76.50 |

Table 8 shows the results for the *PRImA* dataset using the MiT-B0 and MNV3sm backbones. We observe that a grid size of 18 achieves the best performance for both encoders, attaining the highest IoU and mAP scores. Smaller grid sizes result in lower IoU and mAP scores due to excessive splitting of the isothetic covers. Likewise, we find an appropriate grid size for *DocLayNet* and *PubLayNet* as 8 and 10, respectively, by following the same procedure.

The proposed method relies on selecting an appropriate grid size for each dataset, which may vary depending on the characteristics of the document images, such as resolution, density, and diversity of layout elements. Therefore, we conduct hyperparameter tuning to determine the grid size that optimizes performance on the validation set. Empirically, we have observed that the optimal grid size is approximately $\frac{1}{100}^{th}$ of the minimum dimension of the image. Based on this observation, we set the grid size to be $\frac{1}{100}^{th}$ of the minimum dimension of the image in all of our experiments.

### 4.3.5 Varying image size

We next evaluate the effect of different image sizes on the segmentation performance of AutoDocSegmenter using the *PRImA* dataset. We use MiT-B0 and MNV3sm as encoder networks to extract features from the input images, and compare the results using four metrics: IoU, mAP, AP@50, and AP@75. Table 9 summarizes our findings. Table 9 reveals that the segmentation performance is influenced by the image size, and that the optimal size varies depending on the encoder network. For both encoders, the highest resolution of $1024 \times 1024$ pixels achieves the best results for most metrics, indicating that more details and features of the document layout are captured and exploited by the encoders. However, this also increases the memory and computational demands of the model. In this work, we consider image resolution of $256 \times 256$ for our experiments (except in Table 9) as this setting provides a reasonable trade-off between generalization performance and efficiency for resource constrained scenarios (e.g., mobile applications).

### 4.3.6 Performance on document object types

We also analyse the performance of AutoDocSegmenter-U-MiT-B0 and AutoDocSegmenter-U-MNV3sm on various document objects, such as, paragraphs, figures and tables. The training phase is self-supervised and

Table 10: mIoU scores of AutoDocSegmenter-U on paragraphs, figures and tables (*PRImA* dataset).

| Model | Paragraph | Figure | Table |
|---|---|---|---|
| AutoDocSegmenter-U-MiT-B0 | 86.40 | 60.67 | 72.96 |
| AutoDocSegmenter-U-MNV3sm | 85.72 | 65.90 | 71.88 |

Table 11: Comparing different AutoDocSegmenter approaches on *PRImA* and $M^6$-*Doc* datasets.

| Model | *PRImA* | $M^6$-*Doc* |
|---|---|---|
| AutoDocSegmenter-I | 82.40 | 71.50 |
| AutoDocSegmenter-I-MiT-B0 | 89.57 | 75.22 |
| AutoDocSegmenter-I-MNV3sm | 85.46 | 75.25 |
| AutoDocSegmenter-U | 82.20 | 72.90 |
| AutoDocSegmenter-U-MiT-B0 | **89.65** | **75.56** |
| AutoDocSegmenter-U-MNV3sm | 87.76 | 75.24 |

our model predicts only the segmentation mask for each document image without labeling the underlying objects. The *PRImA* dataset, on the other hand, has labels corresponding to each object in the ground truth annotation. Hence, we perform connected component analysis on the masks generated by our approach and label each component based on the maximum overlap of the component with the ground truth. We report the average mIoU scores obtained for each object in Table 10. We observe that the mIoU scores for paragraphs are high followed by tables and figures. This is because the mask of figures in *PRImA* dataset have diverse and complex shapes compared to paragraphs and tables. Moreover, for some complex document images, the ground truth does not cover all the objects while our model is able to detect them (e.g., shown in A.3).

### 4.3.7 Comparing variants of AutoDocSegmenter

To study the effectiveness of Algorithm 1 and the self-supervised training stage, we evaluate the performance of the unsupervised algorithms AutoDocSegmenter-I and AutoDocSegmenter-U and their self-supervised variants with MiT-B0 and MNV3sm backbones on *PRImA* and $M^6$-*Doc*. These two datasets closely resemble real world documents consisting of complex layouts with varying background textures. The self-supervised learning methods are trained *PubLayNet* (i.e., we follow the generalized setting 1 discussed in Section 4.2.2). Table 11 reports the performance of all the six methods. We observe that AutoDocSegmenter-U achieves comparable or better performance than AutoDocSegmenter-I. Similarly, the self-supervised AutoDocSegmenter-U-MiT-B0 and AutoDocSegmenter-U-MNV3sm retain this advantage over their self-supervised AutoDocSegmenter-I counterparts.

## 5 Conclusions

The scarcity of large-scale datasets with precise polygonal annotations of document objects remains a significant obstacle in developing document segmentation models. Our work introduces AutoDocSegmenter, a self-supervised approach utilizing isothetic covers as pseudo masks to train an encoder-decoder model. AutoDocSegmenter effectively handles diverse and complex document layouts, producing accurate segmentation masks and generalizing well to unseen datasets. Additionally, it supports lightweight encoder architectures, offering notable parameter and computational efficiency. Overall, we develop an end-to-end self-supervised document segmentation method for real-world applications.

**Limitations and Future Works**: Although AutoDocSegmenter can handle complex document layouts, it only segregates the foreground information from the background and segments various objects present in the document as binary masks. As it is a self-supervised approach without any supervised fine-tuning, our method cannot identify or classify the objects into different categories. We plan to investigate this unsupervised classification problem as a future work.

## 6 Broader Impact Statement

We present a self-supervised technique of document segmentation which can handle complex layouts efficiently and can be integrated with both lightweight and heavyweight encoder architectures based on the available resources. Overall, our work reduces the dependency on human annotations for segmentation task, which is a tedious task. To the best of our knowledge, this work does not pose any negative societal impact.

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

# A Appendix

## A.1 Data Description

We evaluate our method on four popular document segmentation datasets: *PRImA* (Antonacopoulos et al., 2009), *DocLayNet* (Pfitzmann et al., 2022), *PubLayNet* (Zhong et al., 2019), and $M^6$-*Doc* (Cheng et al., 2023).

*PRImA* dataset (Antonacopoulos et al., 2009) has 382 training and 96 testing samples, respectively with the image dimensions varying between 2500 to 3500. The annotations of this dataset provide polygonal boundaries for each entity present in the document.

However, for *DocLayNet* (Pfitzmann et al., 2022) and *PubLayNet* (Zhong et al., 2019), the bounding boxes are rectangular in shape. *DocLayNet* has 69375 training and 6489 testing images, respectively. All the images are of fixed dimensions of $1025 \times 1025$. On the other hand *PubLayNet*, is a large-scale dataset with image dimensions ranging from 750 to 950 and contain 335703 training and 11245 testing samples, respectively.

$M^6$-*Doc* (Cheng et al., 2023) is a recently released dataset which introduces diverse set of documents including scanned, photographed and PDFs of scientific articles, books, magazines, newspapers and notes in English and Chinese languages. It has been used for testing the effectiveness of our generalised model. The test set comprises of 2724 images of dimensions roughly varying from $2034 \times 2877$ to $2034 \times 2916$.

## A.2 Isothetic Covers Algorithm

The isothetic covers follow a row-major traversal of the document to outline the regions where objects are present based on convex hull algorithm. The convexity of the point determines the type and angle associated to define the direction of movement. A detailed algorithm is presented as follows:

---
**Algorithm 2** Path Traversal of Isothetic covers

---
$g_1$ = Top-left grid occupied
$g_2$ = Top-right grid occupied
$g_3$ = Bottom-left grid occupied
$g_4$ = Bottom-right grid occupied
$t$ = Type of vertex, $t_a$ = Angle of vertex
**if** $(g_1$ & $!(g_2 \& g_3 \& g_4))$ || $(g_2$ & $!(g_1 \& g_3 \& g_4))$ || $(g_3$ & $!(g_1 \& g_2 \& g_4))$ || $(g_4$ & $!(g_1 \& g_2 \& g_3))$ **then**
    $t \leftarrow 1$
    $t_a \leftarrow 90°$
**else if** $(g_1 \& g_2$ & $!(g_3 \& g_4))$ || $(g_3 \& g_4$ & $!(g_1 \& g_2))$ **then**
    $t \leftarrow 0$
    $t_a \leftarrow 180°$
**else if** $(g_1 \& g_4$ & $!(g_2 \& g_3))$ || $(g_2 \& g_3$ & $!(g_1 \& g_4))$ **then**
    $t \leftarrow -1$
    $t_a \leftarrow 270°$
**else if** $(g_1 \& g_2 \& g_3$ & $!g_4)$ || $(g_1 \& g_2 \& g_4$ & $!g_3)$ || $(g_2 \& g_3 \& g_4$ & $!g_1)$ || $(g_1 \& g_3 \& g_4$ & $!g_2)$ **then**
    $t \leftarrow -1$
    $t_a \leftarrow 270°$
**else if** $(g_1 \& g_2 \& g_3 \& g_4)$ || $!(g_1 \& g_2 \& g_3 \& g_4)$ **then**
**continue**
**end if**

---

The type of grid and angle associated with it depends on the occupancy of the four adjacent grids. When either one or three of the grids are occupied with text, image, etc., we denote the type, $t = 1$ and $t_a = 90°$. Therefore, the direction of movement takes a 90° turn and traverses along the boundary of the occupied grid. Similarly, the direction of movement remains the same when two adjacent grids are occupied. There is no effect on the traversal when all or none of the grids are empty. We show more examples of AutoDocSegmenter-U from *DocLayNet* dataset in Fig. 8.

## A.3 Comparing AutoDocSegmenter-U with ground truth masks

We show the difference between groundtruth masks and the masks generated using AutoDocSegmenter-U in Fig. 9 for *PubLayNet* dataset. *PubLayNet* consists of clean images with simple layout. Hence, it is easier to compare the masks generated using human annotation with the isothetic covers.

We observe that for documents with text-only format or sequential paragraphs, the AutoDocSegmenter-U masks are identical to that of the given annotations. However, for tables, instead of a rectangular box denoting the entire table, AutoDocSegmenter-U precisely outline each row and column using polygonal boundaries. Similar results are observed for images as well.

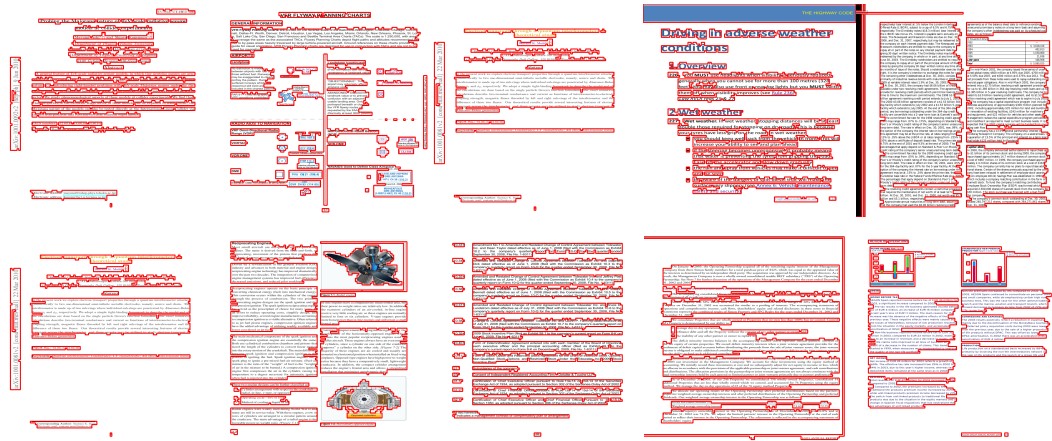

Figure 8: Examples of AutoDocSegmenter-U from *DocLayNet* dataset.

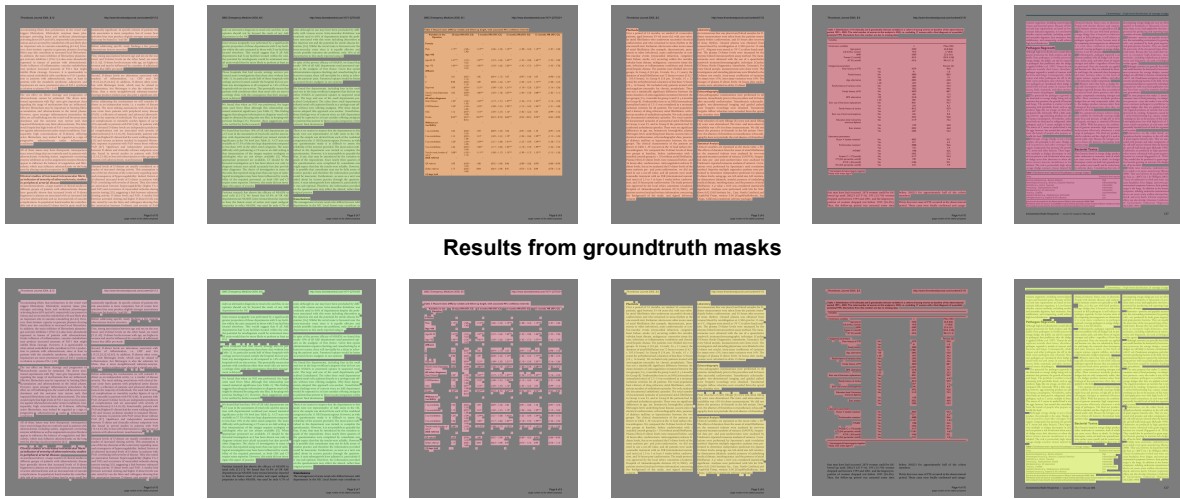

Figure 9: Comparing the ground truth annotations with our AutoDocSegmenter-U masks for *PubLayNet* dataset.

As our proposed learning method is trained using AutoDocSegmenter-U masks, the model learns to mimic the characteristics of polygonal covers and generates similar predicted masks for various objects present in the document. Most complex layout images are annotated using rectangular boxes or left non-annotated in some cases. As AutoDocSegmenter-U is independent of the given groundtruth, it is able to detect complex shaped objects present in the documents. Fig. 10 shows an example of our method applied to a magazine image, where we can see that our network can segment the figure with reasonable precision, even though the training dataset does not contain any tight annotations for the document objects.

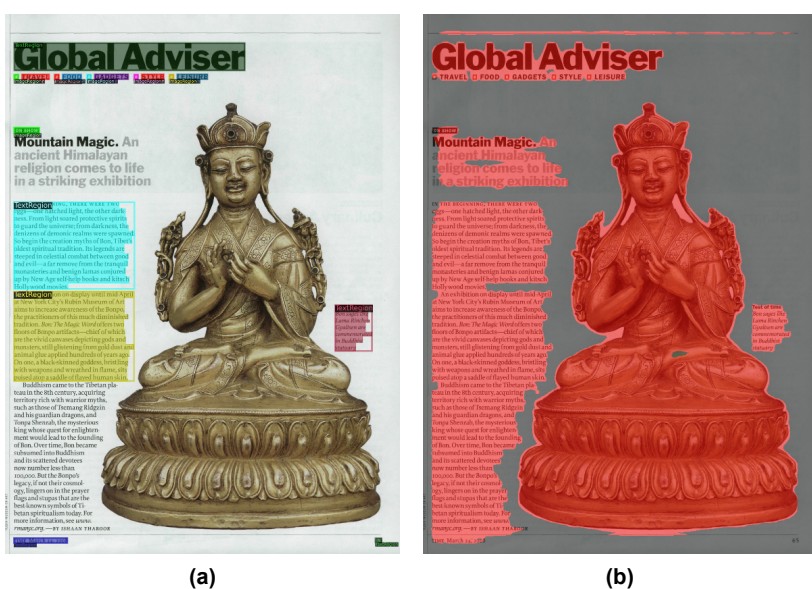

**(a)** **(b)**

Figure 10: Result of a typical magazine image from *PRImA* dataset using our learning technique. (a) The real annotation provided for the document. (b) The predicted mask generated using our proposed learning technique with AutoDocSegmenter-MNV3sm. Our proposed method have been able to outline the complex boundary of the image while it is kept non-annotated in the given dataset.

