# OpenReview forum: "AutoDocSegmenter: A Geometric Approach towards Self-Supervised Document Segmentation"
_TMLR — Accepted by TMLR_

### Review · Reviewer_pYgA · 2024-10-22

**Summary Of Contributions:**

This paper proposes a novel self-supervised approach to the task of document segmentation. There are two main novel contributions:
1. A proposed algorithm for unsupervised generation of pseudo-masks for further training of segmentation model. The algorithm is based on well-known approach for construction of isothetic covers. The novelty lies in that the authors propose to build two sets of isothetic covers for each documents using different binarisation strategies, and then merge two obtained covers;
2. An idea of training an end-to-end document segmentation model based on unsupervisely collected pseudo-masks using algorithm from (1).

**Audience:**

Yes

**Broader Impact Concerns:**

No concerns

**Claims And Evidence:**

No

**Requested Changes:**

Clearer explanation of what algorithms AutoDocSegmenter-I and AutoDocSegmenter-U are and the difference between them. Add motivation and justification for training a segmenting neural network on top of pseudo-masks generated by isothetic covers algorithm.

**Strengths And Weaknesses:**

Strengths of the paper include:
- Clear explanation of the proposed algorithm for unsupervised generation of pseudo-masks;
- Good experimental section with approach tested on sufficient number of datasets and in both in-domain and out-of-domain settings. Many previous approaches are used to compare the proposed method to;
- Good experimental results both in terms of metrics and visual quality;
- Decent ablation with many backbones and hyperparameters tested. Also it is good to see comparisons of the masks generated by the algorithm to the ground-truth ones with the discussion of fail modes.

Elements that require attention:

It is not clear what the difference between AutoDocSegmenter-I and  AutoDocSegmenter-U is. One can assume from the text that AutoDocSegmenter-I is a version of approach which generates masks based only on the proposed approach of constructing and merging isothetic covers, i.e. without training a neural network-based segment, while AutoDocSegmenter-U is a version of approach which uses a neural network segmentor trained on pseudo-masks generated by AutoDocSegmenter-I. If that is true, it is unclear why there are still two versions of AutoDocSegmenter-I approach in Tables 2 and 3, i.e. AutoDocSegmenter-I-MiT-B0 and AutoDocSegmenter-I-MNV3sm, as  MiT-B0 and MNV3sm refer to different backbones for the neural network segmentor. If AutoDocSegmenter-I is not that, then it is unclear what it is (and what the difference between AutoDocSegmenter-I and AutoDocSegmenter-U is), and another question arises: do we actually need to train any segmentation model on top of pseudo-masks generated by proposed isothetic covers algorithm? Will that be any better than just using proposed isothetic covers algorithm for any new document? What is a motivation behind training a network on top of generated pseudo-masks?

---

### Review · Reviewer_1HGU · 2024-10-26

**Summary Of Contributions:**

Labeling documents is quite a costly process, so most large-scale datasets provide simplified annotations, which compromises the performance of models trained on them. This paper proposes an automatic algorithm to generate pseudo polygonal masks capturing the complex shapes and layouts of diverse document layouts and objects. The algorithm improves on a previous grid-based approach by merging isothetic masks generated from two types of binarized images, each derived from global and local thresholding, respectively. Additionaly, the authors train an encoder-decoder segmentation network using the generated pseudo polygonal masks. The resulting model shows competitive performance on several document segmentation benchmarks and generalizes well to unseen document layouts. The framework is also adaptable, supporting a range of encoder architectures, including both convnet and transformer based models.

**Audience:**

Yes

**Broader Impact Concerns:**

I don't believe there are ethical concerns about this paper. Please discuss the limitation of the proposed method and add the Broader Impact Statement section in the final draft.

**Claims And Evidence:**

No

**Requested Changes:**

Please address the above weaknesses and the following minor points:
- In Algorithm 1, what does S_g == S_l exactly mean? and Pg || Pl means applying the logical or operation?
- In Section 3.2, the value "0" for the dice loss function indicates complete similarity, not "1". Also, how can the loss in Equation (1) become 1?
- In Figure 6, the captions for (a) and (b) should be swapped.
- The authors said they fixed the grid size to be approximately 1/100-th of the minimum dimension of the images for a particular dataset in 4.2.1 while they also said they performed hyper parameter tuning for finding the optimal grid size. Which is correct?

**Strengths And Weaknesses:**

Strengths
- The proposed method can be used in various document-related applications, in diverse real-world scenarios.
- The empirical results are quite good on several benchmarks, and the authors provide several ablation studies to analyze the characteristics of the proposed method.

Weaknesses
- The main contributions of this work are two folds: an automatic pseudo mask generation algorithm, and an encoder-decoder document segmentation network trained using the generated pseudo masks. However, the quality of pseudo mask itself is never directly evaluated in the paper, except some examples in Figure 8. All the experiments evaluate the segmentation model. Please provide more evaluation results demonstrating the high quality of pseudo masks. For example, it would be great to compute segmentation metrics by directly comparing the pseudo masks and ground-truth masks.
- The authors stated in Section 1 that SelfDocSeg is sensitive to hyperparameters such as the kernel size and shape for erosion. However, according to Table 5, the proposed method is also quite sensitive to the grid size, which reduces the robustness of the proposed method.

---

### Review · Reviewer_v7LM · 2024-10-30

**Summary Of Contributions:**

The paper presents AutoDocSegmenter, a self-supervised framework that effectively segments documents into meaningful regions, without relying on labeled data. It introduces an unsupervised pseudo-mask generation procedure to create isothetic covers for self-supervised learning. Compared to prior work, it does not depend on a two-stage pipeline where the second requires annotated data, but solely relies on self-supervision to work in within-domain and cross-domain settings. In both evaluation scenarios, the proposed method achieves impressive results.

**Audience:**

Yes

**Broader Impact Concerns:**

No concerns.

**Claims And Evidence:**

Yes

**Requested Changes:**

- Add number of parameters, pre-train and fine-tune samples for methods in the main tables.
- Ablate merging techniques and binarization strategy.
- Analyse the performance of the model on specific object types.
- (minor) There are a couple of typos in the text. A common one regards the use of the "pseudo" term, which is used inconsistently, e.g., "pseudo-masks" and "pseudo masks", etc.

**Strengths And Weaknesses:**

Strengths:

- The proposed method is the first full self-supervised approach for the task, exploiting a traditional technique as supervision instead of human annotations.
- The method achieves state-of-the-art results in document segmentation when compared with many baseline methods in both within-domain and cross-domain settings.
- The method is independent of architectural choices and can be used on any encoder backbone.
- Not only is data efficient, but AudoDocSegmenter is also compute efficient both at train and at test time due to its use of small architectures

Weaknesses:

- Since baselines differs in backbone and training recipes, it could be beneficial to report number of parameters for each method and, if possible, the train size for the pre-training and fine-tuning. Aside from giving more context for comparison, it would also strengthen the claim that the proposed method is efficient.
- Missing ablation regarding different merging strategies for local and global thresholding, e.g., use different overlap thresholds, using weighted average, etc.
- Missing ablation on binarization techniques, i.e., replace Otsu's method with other approaches
- Analysis on the model's performance on specific document object types, i.e., performance on tables, figures, paragraphs, to demonstrate where the approach improves more.

---

### Decision · Action_Editor_JnEY · 2024-11-29

**Recommendation:** Accept with minor revision

**Comment:**

All the reviewers really appreciate this paper.

One reviewer proposes to comment on an ablation discussed with the authors. This is encouraged.

Apart from the paper the paper is ready from publication.

**Audience:**

Yes.

**Claims And Evidence:**

Yes.